# Structural Insights into the Phosphorylation-Enhanced Deubiquitinating Activity of UCHL3 and Ubiquitin Chain Cleavage Preference Analysis

**DOI:** 10.3390/ijms231810789

**Published:** 2022-09-15

**Authors:** Yujing Ren, Beiming Yu, Lihui Zhou, Feng Wang, Yanfeng Wang

**Affiliations:** Key Laboratory of Molecular Medicine and Biotherapy, School of Life Science, Beijing Institute of Technology, Beijing 100081, China

**Keywords:** UCHL3, simulated phosphorylation, structure, ubiquitin chain cleavage, regulation mechanism

## Abstract

Ubiquitin C-terminal hydrolase-L3 (UCHL3), an important member of the ubiquitin C-terminal hydrolase family, is involved in DNA repair and cancer development. UCHL3 can cleave only complexes of monoubiquitin and its conjugates, such as Ub-AMC, His, or small ubiquitin-like modifier, but not polyubiquitin chains. Phosphorylation of Ser75 promotes the cleavage activity of UCHL3 toward poly-ubiquitin chains in vivo, but biochemical evidence in vitro is still lacking. Here, we first analyzed the structure of simulated phosphorylated UCHL3^S75E^ and the complex of UCHL3^S75E^ with Ub-PA and preliminarily explained the structural mechanism of phosphorylation-enhanced UCHL3 deubiquitinating activity. Additionally, the cleavage activity of UCHL3 toward different types of synthesized poly-ubiquitin chains in vitro was tested. The results showed that purified UCHL3^S75E^ enhanced the cleavage activity toward Ub-AMC compared to UCHL3^WT^. Meanwhile, UCHL3^S75E^ and UCHL3^WT^ did not show any cleavage activity for different types of di-ubiquitin and tri-ubiquitin chains. However, UCHL3 could hydrolyze the K48 tetra-ubiquitin chain, providing compelling in vitro evidence confirming previous in vivo results. Thus, this study shows that UCHL3 can hydrolyze and has a cleavage preference for polyubiquitin chains, which expands our understanding of the phosphorylation regulation of UCHL3 and lays a foundation for further elucidation of its physiological role.

## 1. Introduction

Ubiquitination is a ubiquitous post-translational modification that plays important roles in various life processes, including protein degradation, signal transduction, and DNA damage repair [1,2,3,4,5]. Ubiquitination refers to the covalent attachment of ubiquitin (Ub), a 76-residue polypeptide, to target proteins through a sophisticated three-step enzymatic cascade reaction in which Ub is activated by E1(Ub-activating enzyme) and binds to E2 (Ub-conjugating enzyme), leading to the transfer of Ub from E2 to the target protein through E3 (Ub ligase) [6,7,8]. Ub itself can be ubiquitinated via its seven lysine residues or Met1 to form different types of poly-ubiquitin chains, including the K6-, K11-, K27-, K29-, K33-, K48-, and K63-linked chains and the M1 linear Ub chain [9], thus conferring diverse functions on their linked protein substrates [5,10,11,12]. Different ubiquitin chains function differently depending on their length, type, and the connected substrates of ubiquitin chains [7,13]. So far, mono-ubiquitination, K48 linkage ubiquitination, and K63 linkage ubiquitination of proteins have been studied extensively [14]. For example, K48 ubiquitin chains usually mediate the ubiquitin-26S proteasome system and are involved in the degradation of proteins, whereas K63 ubiquitin chains are associated with intracellular transport, various types of signal transduction, and other non-degradation pathways, such as the inflammatory nuclear factor (NF)-κB signaling and the lysosomal degradation pathway [5,15,16,17,18].

Deubiquitination is the inverse process of ubiquitination and is performed by deubiquitinases (DUBs). DUBs can trim Ub molecules from target proteins and regulate the dynamic balance of protein ubiquitination [19,20]. The human genome encodes several types of DUBs, which can be divided into seven subfamilies [21], including ubiquitin-specific processing proteases (USPs), ovarian tumor proteases (OTUs), ubiquitin carboxy-terminal hydrolases (UCHs), JAD1/PAD/MPN domain containing metallo-enzymes (JAMM), Machado–Joseph disease-related enzymes (MJD), and motif-interacting with ubiquitin-containing novel DUB (MINDY) and zinc finger-containing ubiquitin peptidase 1 (ZUP1) [21,22]. Different families of DUBs have different cleavage specificities for the ubiquitin chains. For example, most DUBs of the JAMM family specifically cleave K63 ubiquitin chains, whereas DUBs of the MINDY family tend to specifically cleave K48 ubiquitin chains. CYLD specifically cleaves the K63 and Met ubiquitin chains, playing a role as a tumor suppressor in the NF-κB signaling pathway. USP30 can preferentially cleave K6 polyubiquitin chains and play an important role in mitochondrial autophagy [16,23,24,25,26,27,28,29]. Therefore, DUBs are associated with many diseases and have been identified as potential therapeutic targets for cancer and neurodegenerative diseases [30,31].

DUBs are strictly regulated to ensure precise biological functions. Post-translational modifications (PTMs), such as phosphorylation, are important regulatory mechanisms of DUBs [32,33,34]. Currently, emerging structural evidence has revealed the regulatory mechanism of DUB phosphorylation [33,34]. First, phosphorylation of a single site can activate DUBs, leading to conformational changes after the phosphorylation of DUBs binds to the substrate. For example, DUBA, a member of the OTU family, can be activated when Ser177 is phosphorylated by CKII kinase and undergoes a conformational change near the active site for Ub to enter instead of the mimic phosphorylation [35,36,37,38]. Second, the phosphorylation of a single site directly activates catalytic activity and changes the conformation of DUBs. USP14, purified from E. coli, is inactive and auto-inhibits, but can be activated when bound to the proteasome [39,40,41,42]. A study has revealed that USP14 can also be activated by AKT-induced phosphorylation or simulated phosphorylation of USP14 at Ser432 and may open the pathway between USP14 and the substrate to release self-inhibition via conformational changes [43]. Phosphorylation can also alter the interactions between DUBs and other proteins. For example, the Ser680 residue of USP8 is phosphorylated during the interphase stage of cell division, which enables USP8 to bind to the 14-3-3 protein. This binding in turn inhibits the catalytic activity of USP8 [44,45,46]. USP8 may undergo a conformational change from an active state to an inhibitory state based on the structure of the 14-3-3 protein that binds to phosphorylated USP8 [47]. Therefore, phosphorylation of DUBs plays an important role in regulating their physiological processes of DUB involved.

UCHL3, an important member of the UCH family, plays an important role in the occurrence and development of cancer, DNA damage repair, osteoblast differentiation, and other diseases and life processes [48,49,50,51,52]. Although the UCH family is the first DUBs subfamily to be described, research on its selectivity to ubiquitin chains is still vague [53]. To date, UCHL3 has been considered to be able to cleave only complexes of monoubiquitin and its conjugates, such as Ubiquitin-7-Amino-4-methylcoumarin (Ub-AMC), His, or SUMO, but not polyubiquitin chains [54,55]. Structural studies have shown that UCHL3 can recognize and bind K27 dimer-Ub (K27di-Ub), making the structure of K27di-Ub more extended from the compact conformation, so that the isopeptide bond between di-Ub changes from an embedded to an open state [56,57,58]. However, UCHL3 did not show activity of hydrolyze K27 di-Ub in vitro. Moreover, some studies have shown that the length of the crossover loop of the UCHL3 structure determines its hydrolytic activity for ubiquitin chains. For example, when the length of the crossover loop of UCHL3 is extended, UCHL3 can hydrolyze the K48 di-Ub chain in vitro [54,59,60,61,62]. However, the cleavage activity of UCHL3 toward a certain ubiquitin chain remains unclear and requires further study.

In 2016, Luo et al. first found that phosphorylated UCHL3 at Ser75 can cleave single or polyubiquitin chains on the substrate RAD51 and play an important role in DNA damage repair and tumor occurrence. Phosphorylated UCHL3 showed stronger ubiquitin chain cleavage activity against RAD51 in vivo [52]. However, the chain cleavage type and specificity of UCHL3 remains unclear. Furthermore, the mechanism by which phosphorylation confers the ability of UCHL3 to cleave polyubiquitin chains in vitro remains unclear. Ser75 is not conserved in other UCH family members, suggesting that phosphorylation regulation at Ser75 is unique to UCHL3.

In this study, Ser75 on UCHL3 was mutated to Glu to achieve a simulated phosphorylation modification. Wild-type UCHL3 (UCHL3^WT^) and simulated phosphorylated UCHL3 (UCHL3^S75E^) were recombinantly expressed and purified using affinity chromatography, ion exchange chromatography, and gel filtration chromatography. We first analyzed the structure of UCHL3^S75E^ and the complex of UCHL3^S75E^ with its substrate, Ubiquitin Propargylamide (Ub-PA), and explained the mechanism of phosphorylation-promoted activity of UCHL3. Finally, the cleavage activity of UCHL3^S75E^ and UCHL3^WT^ on different types of synthesized ubiquitin chains was studied in vitro, and it was found that UCHL3 preferentially cleaved activity toward K48 tetramer-Ub (K48 tetra-Ub) chains. These studies are expected to lay a foundation for understanding the molecular mechanism of phosphorylation modification and the ubiquitin chain cleavage preference of UCHL3.

## 2. Results

### 2.1. Cloning, Expression, and Purification of UCHL3^S75E^ from E. coli

The full-length human UCHL3 contained 230 amino acids, which were amplified by polymerase chain reaction (PCR) and ligated into the pET28b vector, yielding an expression construct with an N-terminal His6 tag.

The QuikChange Site-Directed Mutagenesis Kit was used to construct the plasmid UCHL3^S75E^, and then UCHL3^S75E^ was overexpressed and purified. The plasmid and protein purification results for UCHL3^S75E^ are shown in Appendix A.

### 2.2. Crystal Structure of UCHL3^S75E^

UCHL3^S75E^ was constructed by site-directed mutagenesis, overexpressed in Escherichia coli (E. coli) strain *BL21* (DE3) using the vector pET-28a (GE Healthcare), and purified to homogeneity. To explore whether phosphorylation changes the deubiquitinating activity of UCHL3 by directly causing its conformational changes, crystallization trials for UCHL3^S75E^ were carried out. The structure of UCHL3^S75E^ was successfully determined by molecular replacement using an atomic model of human UCHL3^WT^ (PDB ID:1UCH [61]). The X-ray structure of UCHL3^S75E^ was refined to a 2.5 Å resolution (Figure 1A); the data collection and refinement statistics are given in Table 1. The structure of UCHL3^S75E^ was similar to that of UCHL3^WT^, with an RMSD of 0.273 Å. UCHL3^S75E^ caused an outward shift of amino acids at positions 74, 75, and 76, resulting in a 1.2 Å outward shift on the α3 helix compared to that in UCHL3^WT^ (Figure 1B). In addition, after phosphorylation, residues 91 and 92 from the loop connecting the active site of catalytic residue Cys95 have an outward shift and may contribute to the exposure of the active site and recognition of the substrate (Figure 1C).

### 2.3. Crystal Structure of UCHL3^S75E^ in Complex with Ub-PA

The molecular mechanism of phosphorylation-enhanced UCHL3 deubiquitination activity has not been well explained due to the lack of obvious conformational changes. We propose that phosphorylation modification may cause a conformational change in UCHL3 after binding to the ubiquitin substrate, similar to DUBA [38], a member of the OTU family. Therefore, the co-crystal structure of UCHL3^S75E^ bound to Ub-PA (UCHL3^S75E^-Ub-PA) at high resolution (2.5 Å; Figure 2A) was obtained and successfully solved by molecular replacement using the atomic model of human UCHL3^WT^ bound to Ub-VME(UCHL3^WT^-Ub-VME) (PDB ID:1XD3 [62]). Data collection and refinement statistics are presented in Table 1.

We compared the structures of UCHL3^S75E^-Ub-PA with UCHL3^WT^-Ub-VME and found that the simulated phosphorylation modification still resulted in a 3.5 Å outward shift of the α3 helix in which the amino acids at positions 74, 75, and 76 of UCHL3^S75E^-Ub-PA displaced relative to that in UCHL3^WT^-Ub-VME (Figure 2B). In addition, we compared the structure of UCHL3^S75E^ in the UCHL3^S75E^-Ub-PA complex with that of apo UCHLL3^S75E^ and found that their structures were almost similar to each other with an RMSD of 0.472. Specifically, the amino acids at positions 91 and 92 of UCHL3^S75E^-Ub-PA had an outward position change relative to that of apo UCHL3^S75E^, which might make it easier to expose and enhance its deubiquitinating activity (Figure 2C). The amino acids Arg72, Leu73, and Arg 74 are located on the loop at the C-terminus of Ub, which is the region entering the active center of UCHL3. From the structure of UCHL3^WT^-Ub-VME, we can see that Pro8 on UCHL3 interacts with Arg74 on Ub via hydrogen bonding. However, the interaction force becomes weaker after phosphorylation because of the larger distance between the two amino acids, which may increase the width of the region where the substrate enters the active center, allowing the entry of larger substrates. Meanwhile, the hydrophobic interaction between Arg 72 on Ub and Gln 12 on UCHL3 and the hydrogen bond between Ala11 on UCHL3 and Leu73 on Ub become stronger after phosphorylation, which may play a role in the recognition of the substrate by UCHL3 (Figure 2D,E). Therefore, the changes in the interaction force in this region may indicate a change in the recognition and interaction between UCHL3 and the substrate.

### 2.4. Data-Collection and Refinement Statistics of UCHL3^S75E^ and UCHL3^S75E^-Ub-PA

After numerous trials, the crystal structures of UCHL3^S75E^ and UCHL3^S75E^-Ub-PA were successfully solved by molecular replacement using the atomic models of human UCHL3^WT^ (PDB ID:1UCH [61]) and human UCHL3^WT^ bound to Ub-VME(UCHL3^WT^-Ub-VME) (PDB ID:1XD3 [62]). Data collection and refinement statistics are given in Table 1.

### 2.5. Phosphorylation Enhanced the Deubiquitinating Activity of UCHL3

We observed some changes in the structure of UCHL3^S75E^ and the complex of UCHL3^S75E^-Ub-PA and attempted to confirm whether phosphorylation modification can improve the activity of UCHL3 in vitro. A ubiquitin-7-amino-4-methylcoumarin (Ub-AMC) hydrolysis assay was performed as previously described [63]. Compared with UCHL3^WT^, phosphorylation increased Ub-AMC hydrolytic activity by approximately10 times (Figure 3).

### 2.6. Cleavage Activity of UCHL3 to Different Types of Ubiquitin Chains In Vitro

UCHL3 has been considered to cleave only complexes of monoubiquitin and its small conjugates, such as Ub-AMC, Ub-His, and Ub-Sumo [54]. Moreover, the UCH family of DUBs (UCHL1, UCHL3, UCHL5, and BAP1) is reported to be inactive against di-Ub in all linkage types [54]. Luo et al. found that phosphorylated UCHL3 can cleave polyubiquitin chains in vivo, but in vitro evidence is still lacking [52]. To explore the cleavage preference of UCHL3 for different types of ubiquitin chains and whether phosphorylation modification gives UCHL3 the ability to cleave ubiquitin chains in vitro, different linkages (K6, K11, K27, K48, and K63) of di-Ub were synthesized and tested for their activity of UCHL3^WT^ and UCHL3^S75E^. However, both UCHL3^WT^ and UCHL3^S75E^ were unable to hydrolyze di-Ub (Figure 4B–F); the complete gel is shown in Appendix A. Previous structural studies have shown that UCHL3^WT^ can recognize K27 di-Ub and exhibits a more stretched structure of K27 di-Ub, with more exposed isopeptide bonds [55,57,64]. We propose that UCHL3 may recognize K27 di-Ub and cleave longer ubiquitin chains.

We then analyzed the activity of UCHL3 toward longer M1, K48, and K63 trimer-Ub (K63 tri-Ub), which was synthesized based on K27 di-Ub (K27/M1, K27/48, K27/63). However, UCHL3^WT^ and UCHL3^S75E^ were unable to cleave any tri-Ub chains, apart from slight activity against K27/48 tri-Ub at a substrate concentration of 5 uM (Figure 5C–E). Further experiments showed that UCHL3^S75E^ and UCHL3^WT^ could hydrolyze K27/48 tri-Ub to di-Ub when the concentration of K27/48 tri-Ub was 15 uM (Figure 5F); the full gel is shown in Appendix A.

We then synthesized K48 tetra-Ub attached to cyclinB1(cyclinB1-K48 tetra-Ub) as a substrate to test whether UCHL3^WT^ or UCHL3^S75E^ could hydrolyze longer ubiquitin chains (Figure 6A). Sodium dodecyl sulfate-polyacrylamide gel electrophoresis (SDS-PAGE) results showed that both UCHL3^WT^ and UCHL3^S75E^ can cleave cyclinB1-K48 tetra-Ub into several bands, which was verified by mass spectrometry. The red arrow presents cyclinB1 and mono-Ub, as the molecular weight of mono-Ub is similar to that of cyclinB1. The blue arrow represents di-Ub, corresponding to the control sample of K48 di-Ub. Additionally, based on the mass spectrometry analysis and molecular weight, the purple and yellow arrows indicate tetra-Ub and tri-Ub, respectively. Therefore, we concluded that UCHL3^WT^ and UCHL3^S75E^ can hydrolysis cyclinB1-K48 tetra-Ub into cyclinB1, mono-Ub, di-Ub, tri-Ub, and tertra-Ub (Figure 6B).

## 3. Discussion

PTMs of DUBs play important roles in various cellular activities, and phosphorylation is one of the most common modes. According to existing reports, the molecular mechanism of phosphorylation-regulated deubiquitination activity changes in DUBs is mainly divided into three ways. First, phosphorylation modification directly causes conformational changes in DUBs such as USP14; phosphorylation at Ser432 may open the BL2 loop of USP14 to release self-inhibition and promote hydrolysis of Ub substrates [43]. In the second method, after phosphorylation modification, the conformation of DUBs changes only after binding to the substrate, such as the phosphorylated DUBA bound to Ub, which alters its conformation instead of the phosphorylated DUBA [38]. Another way is that phosphorylation recruits a partner to bind with DUBs, such as USP8, which enables USP8 to interact with 14-3-3 proteins, thereby inhibiting its activity [44]. We found that UCHL3^S75E^ had a slight structural change in the α3 helix and a loop near the active center compared to UCHL3^WT^. Therefore, the mechanism of phosphorylation-induced UCHL3 activity changes belong to the conformational changes caused by phosphorylation of the noncatalytic center, and phosphorylated UCHL3 has a greater degree of structural changes when bound to the Ub substrate. However, we did not observe obvious changes in the structures of UCHL3^S75E^ and UCHL3^S75E^-Ub-PA, so as not to provide a clear illustration of the mechanism of phosphorylation-induced activity changes in UCHL3.

We found that UCHL3 had a cleavage preference toward K27/48 tri-Ub and cyclinB1-K48 tetra-Ub chains, but UCHL3^S75E^ did not show enhanced cleavage activity compared to UCHL3^WT^. These results may be due to the difference between the simulated and true phosphorylation by ATM. Similar to DUBA, it is inactive by mimic phosphorylation, whereas it can be activated when it is phosphorylated by CKII kinase; there are structural changes after binding to the Ub substrate instead of apo DUBA [38]. In addition, the polyubiquitin chains on the substrate RAD51 can be cleaved by ATM-phosphorylated UCHL3 instead of simulated phosphorylated UCHL3 in vivo. Nevertheless, these slight structural changes might also provide some clues to explain the phosphorylation-enhanced activity of UCHL3.

Previous studies have analyzed the crystal structure of UCHL3-K27 di-Ub, in which one ubiquitin occupies the S1 site of UCHL3 and the other ubiquitin occupies the S2 site of UCHL3 (Figure 7A) [57]. It is known that the cleavage of ubiquitin chains by DUBs makes the isopeptide bond as close to the active center as possible. Therefore, if UCHL3^S75E^ can cleave polyubiquitin chains, then there should also be an S1′ site. The Ub moiety that binds to the enzymatic S1′ site contributes to Lys, which binds to the carboxyl group contributed by the Ub bound at the S1 site to form an isopeptide bond, penetrating the active center. Therefore, we propose that UCHL3 may use di-Ub as a recognition unit to cleave polyubiquitin chains. That is, ubiquitin occupies the S1 and S2 sites of UCHL3 and then cleaves the isopeptide bond between S1 and S1′ (Figure 7B). The phosphorylation site Ser75 of UCHL3 is located on S1′, and phosphorylation of Ser75 may help S1′ to form the correct conformation and enhance the interaction between ubiquitin chains and UCHL3.

We speculated the mechanism of hydrolysis of polyubiquitin chains by UCHL3 by analyzing the structure of UCHL3^WT^ complexed with K27 di-Ub, UCHL3^S75E^, and UCHL3^S75E^ -Ub-PA. Meanwhile, we found that UCHL3 can cleave K27/48 tri-Ub into di-Ub and mono-Ub and can cleave cyclinB1-K48 tetra-Ub into tri-Ub, di-Ub, and mono-Ub, but it cannot cleave di-Ub. These biochemical assays first prove that UCHL3 has a preference for the K48 polyubiquitin chain and then prove our idea of how UCHL3 hydrolyzes ubiquitin chains. Therefore, analyzing the high-resolution crystal structure of the UCHL3 complex bound to K48 tri-Ub or K48 tetra-Ub may further clarify the molecular mechanism by which UCHL3 recognizes and specifically cleaves K48 linkage ubiquitin chains.

We know that phosphorylated UCHL3 by ATM can deubiquitinate RAD51 in vivo; however, our studies showed that mimetic phosphorylated UCHL3 can only cleave K48 tri-Ub or longer K48 tetra-Ub with a slower rate in vitro. Just like USP13, it can also catalyze the hydrolysis of K63 and K48 poly-Ub chains, but with a very low hydrolyzing rate [65]. The regulation of UCHL3 may be complex and may require the cooperative participation of multiple DUBs rather than a single UCHL3 in vivo. In addition, phosphorylated UCHL3 by ATM may show excellent hydrolytic properties toward K48 ubiquitin chains compared to wild-type UCHL3. On the other hands, in order to further understand the type of ubiquitin chains hydrolyzed by UCHL3 in vivo, we can obtain the ubiquitinated RAD51 in vivo, which can then be investigated by enzyme digestion and mass spectrometry. From these studies, the functional substrates and mechanisms of UCHL3 may be illustrated clearly.

## 4. Materials and Methods

### 4.1. Protein Preparation

His-tagged full-length UCHL3^WT^ was cloned into the pET-28a expression vector. The protein was overexpressed and purified, as previously described [57]. The phosphorylated UCHL3^S75E^ plasmid was generated by PCR using a QuikChange Site-Directed Mutagenesis Kit (TransGen Biotech, Beijing, China), and the protein was overexpressed and purified using the same method.

### 4.2. UCHL3^S75E^ Bound to Ub-PA

Ub-PA was a gift from the Lei Liu laboratory (Tsinghua University, Beijing, China); UCHL3S75E and Ub-PA were incubated in a molar ratio of 2:1 in a 100 uL reaction system containing 25 mM Tris-HCl, pH 7.5, 150 mM NaCl at 4, and 18 °C, respectively; samples were taken at 1, 2, and 4 h, respectively. The complex was analyzed via SDS-PAGE, and excess UCHL3^S75E^ and Ub-PA were removed on a Superdex 75 size-exclusion column run with 10 mM Tris pH 7.5 and 150 mM NaCl, and the final complex was concentrated to 20 mg. mL^−1^. The results for UCHL3^S75E^ bound to Ub-PA are shown in Appendix A.

### 4.3. Crystallization of UCHL3^S75E^-Ub-PA and UCHL3^S75E^

Crystals of UCHL3^S75E^- Ub-PA were grown for a week at 18 °C by the hanging drop method by mixing the UCHL3^S75E^-Ub-PA with an equal volume of reservoir solution containing 25% PEGMME 2000 and 0.1 M Bis-Tris (pH 6.9). Crystals were equilibrated in a cryoprotectant buffer containing reservoir buffer and 20% ethylene glycol (*v/v*) and were flash frozen in a cold nitrogen stream at –170 °C. Crystals of UCHL3^S75E^ were grown for 4–7 days by mixing UCHL3^S75E^ with an equal volume of reservoir solution containing 0.1 M Bis-Tris (pH 8.5), 0.3M MgCl_2_, and 30% PEG 4000 using the hanging drop method; the crystals were equilibrated using the same method as above.

### 4.4. X-ray Data Collection and Structure Determination

Native diffraction datasets for UCHL3^S75E^-Ub-PA and UCHL3^S75E^ were collected on beamline BL17U1 at the Shanghai Synchrotron Radiation Facility and processed using HKL2000. Subsequent processing was performed using programs from the CCP4 suite. The UCHL3^S75E^-Ub-PA and UCHL3^S75E^ structures were solved by molecular replacement using PHASER [66]. Human UCHL3^WT^-Ub-VME (PDB ID:1XD3 [62]) and UCHL3^WT^ (PDB ID:1UCH) were selected as research models for molecular replacement. The structure was refined using PHENIX software [67]. The statistical analysis results of the data are summarized in Table 1.

### 4.5. Ub-AMC Hydrolysis Assay

Ub-AMC was used to measure the deubiquitinating activity of UCHL3^WT^ and UCHL3^S75E^, which were gifts from the Lei Liu laboratory (Tsinghua University, Beijing, China). The reaction mixture contained 50 mM Tris-HCl (pH 7.5), 1 mM EDTA, 1 mg/mL ovalbumin, 5 mM ATP/MgCl2 (freshly prepared), 1 mM DTT (freshly prepared), 0.05% Tween, and 5 nM UCHL3^S75E^, UCHL3^WT^, or 50nM UCHL3^WT^. To start the reaction, 1 μM of Ub-AMC was added to the system. Ub-AMC hydrolysis was measured at Ex355/Em460 on an EnVision plate reader (PerkinElmer). Fluorescence intensity was recorded every 250 s for 50 min or 500 s for 150 min. The experiment was repeated thrice, and the average value was calculated. The result was calculated based on the rate from the start time to the time of starting equilibrium point.

### 4.6. Ubiquitin Chain Cleavage Assays

Different types of ubiquitin chains were obtained from Yiming Li Laboratory [68] (Hefei University of Technology, Hefei, China). These were used to measure the deubiquitinating activity of UCHL3^WT^ and UCHL3^S75E^ in vitro. Di-Ub cleavage reactions containing 5 μM substrate and 3 μM enzyme were conducted in a reaction buffer including 25 mM Tris-HCl (pH 8.0) and 150 mM NaCl at room temperature for different time points. Finally, the hydrolysis activity was analyzed using SDS-PAGE gels. For the tri-Ub and tetra-Ub chain hydrolysis assays, the reaction conditions were the same as above.

## 5. Conclusions

UCHL3 has been designated as an important DUB in various cancers and for DNA repair. A study also found that phosphorylation of Ser75 promotes the cleavage activity of UCHL3 toward poly-ubiquitin chains in vivo. Therefore, it is necessary to describe the mechanism of phosphorylation modification and the preference of Ub chains in detail. In this study, in vitro assay showed that purified UCHL3^S75E^ enhanced the cleavage activity toward Ub-AMC compared to UCHL3^WT^. We first analyzed the structures of the simulated phosphorylation of UCHL3^S75E^ and the complex of UCHL3^S75E^ with Ub-PA. We found that UCHL3^S75E^ has slight structural changes in the α3 helix and the loop near the active center compared to UCHL3^WT^, and it is preliminarily speculated that the exposure of the active center of UCHL3 and the enhanced recognition of substrates are the reasons for the phosphorylation modification to enhance the activity of UCHL3 deubiquitinase. We proved that UCHL3^S75E^ also cannot cleave any type of di-Ub chains; however, UCHL3 tends to cleave longer Ub chains and has a preference for K48-linked Ub chains. These findings expand our understanding of the phosphorylation regulation of UCHL3 and clarify the multicomplex functions of UCHL3.

## Figures and Tables

**Figure 1 ijms-23-10789-f001:**
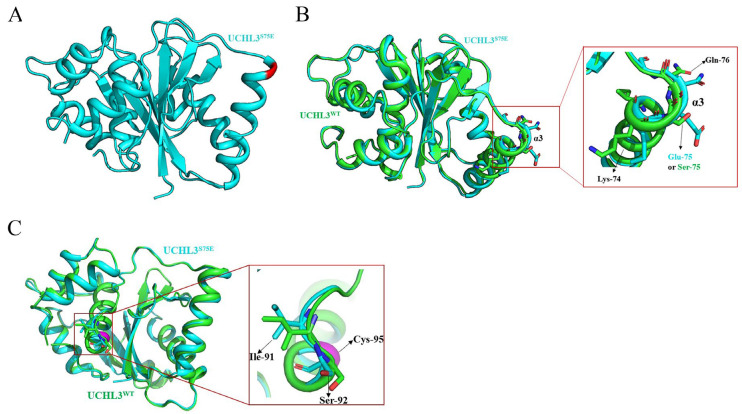
Crystal structure of ubiquitin C-terminal hydrolase-L3 (UCHL3) ^S75E^**.** (**A**) Overview of the structure of UCHL3^S75E^. (**B**) Structural comparison of UCHL3^S75E^ and UCHL3^WT^; the magnified part shows the structural changes of UCHL3^S75E^ and UCHL3^WT^ on α3 helix. (**C**) Structural changes of amino acids near catalytic residue Cys95; magenta indicates the active site of Cys95. In all panels, UCHL3^S75E^ is in cyan, UCHL3^WT^ is in green, and amino acid residues 75 and 95 are in red and magenta, respectively.

**Figure 2 ijms-23-10789-f002:**
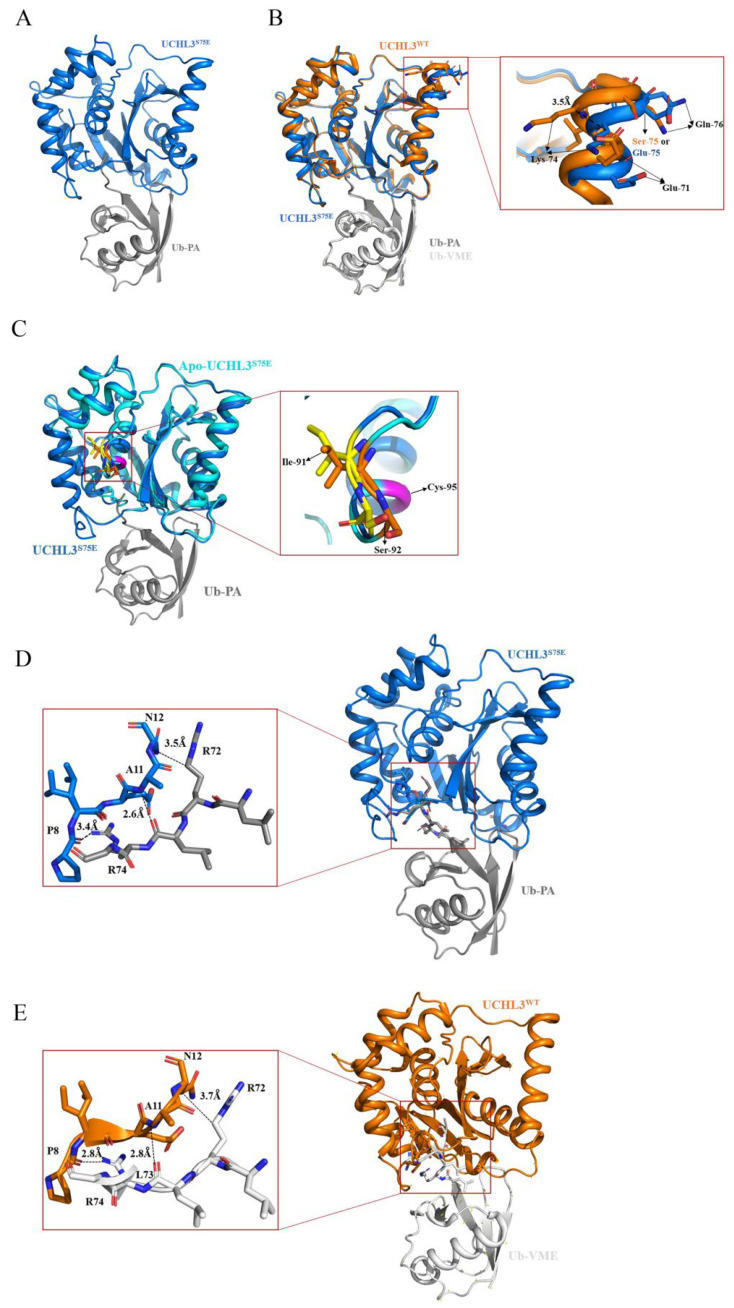
Crystal structure of UCHL3^S75E^ bound to Ub-PA. In all panels, UCHL3^S75E^ is in blue, UCHL3^WT^ is in orange, and Ub-PA and Ub-VME are in gray and white, respectively. (**A**) Overview of the structure of UCHL3^S75E^-Ub-PA in a cartoon representation. (**B**) Structural changes near the phosphorylation site of complex UCHL3^S75E^-Ub-PA. (**C**) Structural changes near the active site between UCHL3^S75E^-Ub-PA and apo’s UCHL3^S75E^; yellow and orange represent amino acid residues Ile91 and Ser92 of UCHL3^S75E^-Ub-PA and UCHL3^S75E^, respectively. (**D**) Interaction of UCHL3^S75E^ with the C terminus of ubiquitin. (**E**) Interaction of UCHL3^WT^ with the C terminus of ubiquitin.

**Figure 3 ijms-23-10789-f003:**
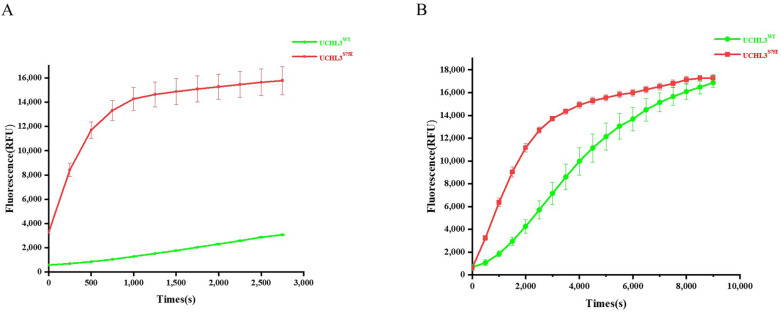
Ub-AMC hydrolysis activity of UCHL3^WT^ and UCHL3^S75E^. UCHL3 and Ub-AMC were used to start the reaction; the fluorescence intensity of Ub-AMC at ex355/em460 was measured and recorded every 250 s for 50min or 500s for150min. The experiment was repeated three times, the average value was calculated, and then ORIGIN was used to process the data. (**A**) Ub-AMC hydrolysis result of 5 nM UCHL3^S75E^ and 5 nm UCHL3^WT^. (**B**) Ub-AMC hydrolysis result of 5 nM UCHL3^S75E^ and 50 nm UCHL3^WT^.

**Figure 4 ijms-23-10789-f004:**
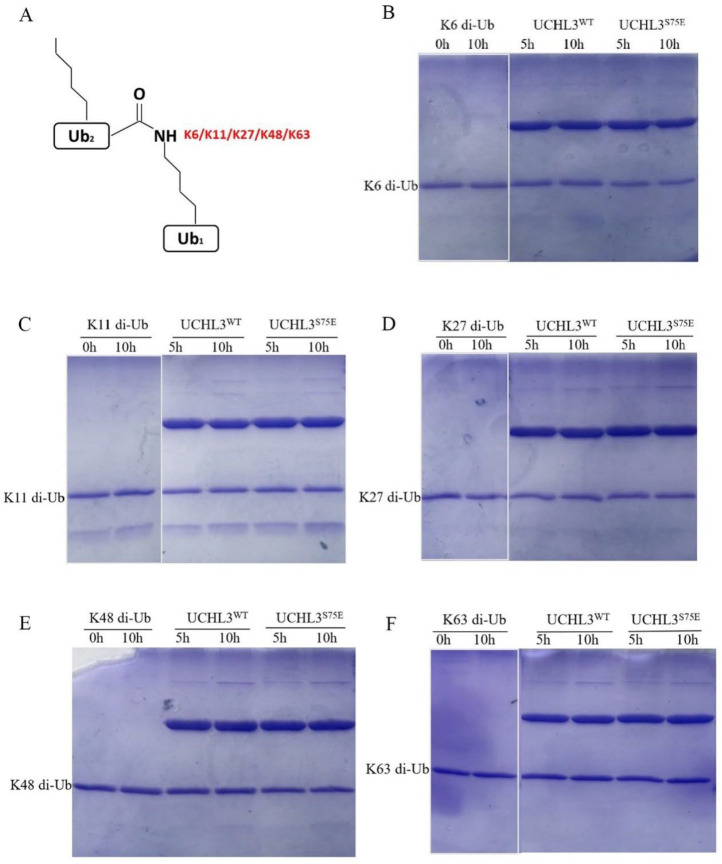
Hydrolysis activity of UCHL3 toward different types of di-Ub. (**A**) Pattern diagrams of different types of synthesized di-Ub (K6, K11, K27, K48, K63). (**B**–**F**) UCHL3 was incubated with different types of di-Ub, and the reaction products were fractionated via sodium dodecyl sulfate-polyacrylamide gel electrophoresis (SDS/PAGE) and visualized by Coomassie Blue staining.

**Figure 5 ijms-23-10789-f005:**
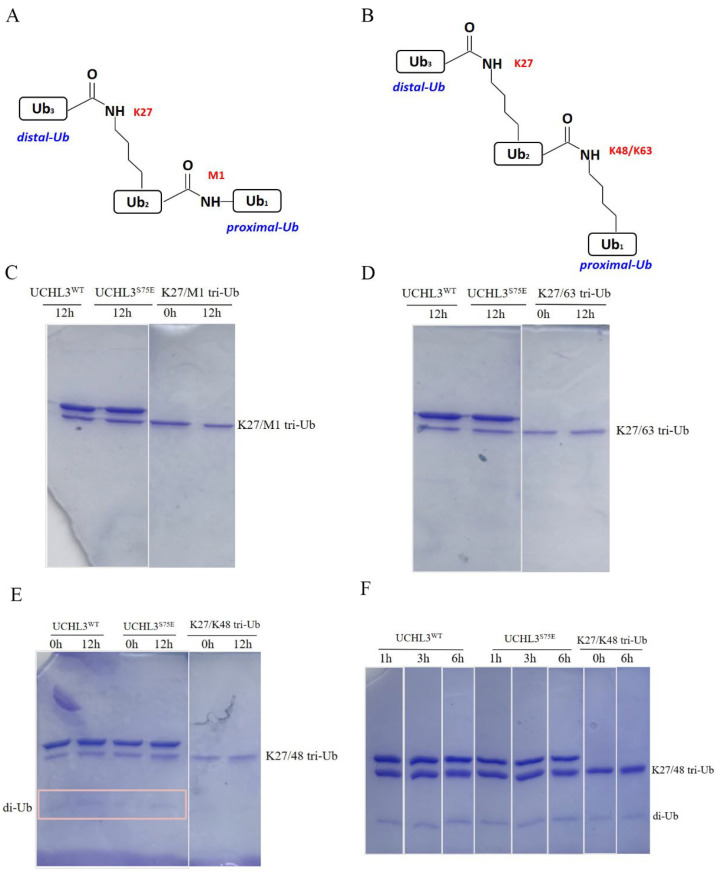
Hydrolysis activity of UCHL3 toward different types of tri-Ub chains. (**A**,**B**) Pattern diagrams of synthesized K27/M1 tri-Ub, K27/63 tri-Ub, and K27/48 tri-Ub. (**C**–**F**) UCHL3^S75E^ or UCHL3^WT^ was incubated with the substrates, respectively, and the reaction products were fractionated via SDS/PAGE and visualized by Coomassie Blue staining. (**E**) Reaction under 3uM UCHL3 and 5uM K27/48 tri-Ub. (**F**) Reaction under 3uM UCHL3 and 15uM K27/48 tri-Ub.

**Figure 6 ijms-23-10789-f006:**
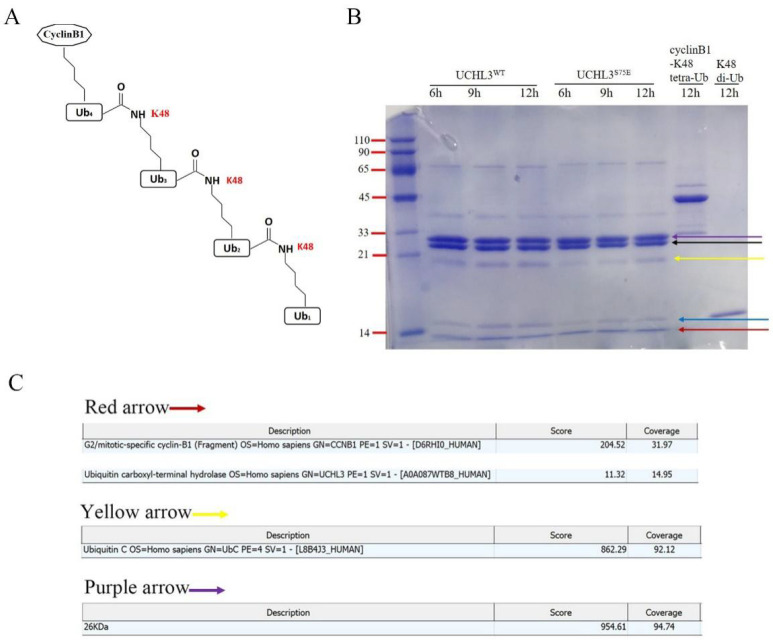
Hydrolysis activity of UCHL3 toward cyclinB1-K48 tetra-Ub. (**A**) Pattern diagram of the substrate of cyclinB1-K48 tetra-Ub. (**B**) UCHL3^S75E^ or UCHL3^WT^ was incubated with the substrate and the reaction products were fractionated by SDS/PAGE and visualized by Coomassie Blue staining; different products are indicated by arrows of different colors. The black arrow indicates UCHL3^WT^ or UCHL3^S75E^. (**C**) Mass spectrometry results of the hydrolysis bands correspond to the arrows in different colors.

**Figure 7 ijms-23-10789-f007:**
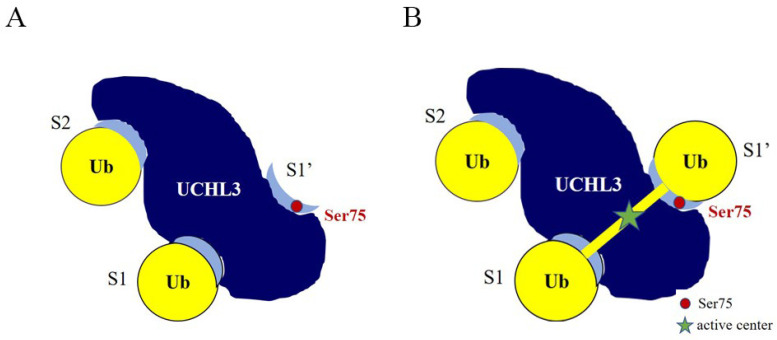
(**A**) Model of the complex of UCHL3 bound to K27 di-Ub. (**B**) Model of phosphorylation-modified UCHL3 hydrolyzing ubiquitin chains.

**Table 1 ijms-23-10789-t001:** Statistics for the highest-resolution shell are shown in parentheses.

	UCHL3^S75E^	UCHL3^S75E^-Ub-PA
Data collection		
Space group	P 21	P 43
Unit-cell parameters	55.163 59.878 80.379 90 90 90	83.052 83.052 178.472 90 90 90
Resolution (Å)	28.06–2.494 (2.583–2.494)	49.06–2.801 (2.901–2.801)
Unique reflections	9508 (865)	29,320 (2810)
Completeness (%)	97.74 (92.01)	98.76 (95.32)
Wilson B-factor	44.59	54.86
Reflections used in refinement	9506 (864)	29,291 (2810)
Reflections used for R-free	487 (51)	1454 (133)
R-work	0.2213 (0.2972)	0.2087 (0.2759)
R-free	0.2876 (0.4247)	0.3130 (0.3899)
Number of non-hydrogen atoms	1634	9701
Macromolecules	1634	9673
Protein residues	205	1236
Ligands	-	28
RMS (bonds)	0.008	0.010
RMS (angles)	0.98	1.20
Ramachandran favored (%)	96.52	91.36
Ramachandran allowed (%)	2.99	7.21
Ramachandran outliers (%)	0.50	1.43
Rotamer outliers (%)	0.00	0.28
Clashscore	11.85	19.03
Average B-factor	50.27	60.29
Macromolecules	50.27	60.28
Ligands	-	63.13

## Data Availability

The data that support the findings of this study are available from the corresponding author upon reasonable request.

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
