# Peer review of "Structural Insights into the Phosphorylation-Enhanced Deubiquitinating Activity of UCHL3 and Ubiquitin Chain Cleavage Preference Analysis"

_ijms, 2022, doi:10.3390/ijms231810789_

Round 1

Reviewer 1 Report

This is an interesting paper on the effect of the S75E mutation of uniquitinase on its structure and activity. According to the Authors, this mutation simulates phosphorylation. By X-ray crystallography, the Authors demonstrated that the mutation results in better exposure of the ubiquitin-binding site and increased hydrolysis of Ub-AMC. The paper is overall well written. The experiments seem to have been performed and their results interpreted correctly and the conclusions are justified. I have the following minor suggestions:

1. The Authors should justify why the selected mutation can simulate phosphorylation of serine. Does any previous study justify this selection? If so, a reference or references should be cited.

2. There are a number of unexplained acronyms, e.g., Ub-AMC, Ub-PA. These should be defined.

3. The sentence in lines 320-322:

"We proved that UCHL3 cannot cleaves any type of di-Ub chain, including UCHL3S75E; however, UCHL3 tends to cleave longer Ub chains and has a preference for K48-linked Ub chains."

is not clear. First, "cannot cleaves" should be replaced by "cannot cleave". Second, the Authors probably mean that "neither UCHL3 nor UCHL3S75E can cleave any type of di-Ub chains". Third, it looks like the Authors did not study the cleaving of poly-Ub chains. Therefore, a reference is required to support the statement that "however, UCHL3 tends to cleave longer Ub chains and has a preference for K48-linked Ub chains."

Author Response

We would like to thank the editors and reviewers for their quick response and valuable advice. We responded to all of the questions point by point and highlighted that in blue in the revised manuscript. Please see the attachment for the detailed response.

Reviewer 2 Report

In their manuscript entitled "Structural Insights into the Phosphorylation-Enhanced Deubiquitinating Activity of UCHL3 and Ubiquitin Chain Cleavage Preference Analysis " the authors provide novel in vitro evidence demonstrating substrate selectivity of the DUB family protein UCHL3. The significance of the study is its insight into the biophysical mechanism of a previously reported but poorly understand in vivo phenomenon. Merits of the manuscript are the quality and presentation of the data and the flow of the text. There are, however, some points of concern that should be addressed prior to publication.

Minor:

1) Company/Software names should be capitalized 

ex: Page 7, line 195 (Figure 3 Legend) Origin should be capitalized 

2) Specific software packages and data analysis methods should be referenced 

ex: PHENIX and PHASER should be cited according to the developers instructions 

3)Similarly, PDB entries used should be accordingly cited with the depositing publication if available

ex: PDB 1XD3 should be cited as 

Structure of the Ubiquitin Hydrolase UCH-L3 Complexed with a Suicide Substrate

Misaghi, S.Galardy, P.J.Meester, W.J.N.Ovaa, H.Ploegh, H.L.Gaudet, R.

(2005) J Biol Chem 280: 1512-1520

  • DOI: 10.1074/jbc.M410770200
  • Primary Citation of Related Structures:  
    1XD3

4)Page 1 lines 43-45 is unclear.  

I assume it is meant to read "such as the inflammatory nuclear factor (NF)kappaB signaling and other non-proteosomal degradation pathways, such as the lysosomal degradation pathway..."

This should be rewritten for clarity.

5) Page 2 lines 76-78 is also unclear and should be rewritten for clarity, particularly the last phrase "which enables USP8 to bind to the 14-3-3 protein and inhibits its activity of USP8..."

Assuming I am interpreting the intended meaning correctly this could be rewritten to read "which enables USP8 to bind to the 14-3-3 protein, inhibiting USP8 activity..."

Major:

I have only one major concern with the manuscript and study in general.  While the use of glutamic acid as a phosphomimetic is generally well accepted, it is important to consider the implications of increasing the carbon side chain length. Aspartic acid is more commonly the preferred mimetic for phosphorylated serine/threonine, while glutamic acid is the preferred mimetic for phosphorylated tyrosine. With the additions carbon-carbon bond the possibility exists steric contributions may confound the results.  As such it would be ideal to test and compare the enzymatic activities of UCHL3 with S75L (non-phosphorylated, bulky) and S75D (phosphomimetic, less bulky).

Author Response

(The authors gave the same response as above.)

Reviewer 3 Report

Several hundred highly regulated ubiquitin ligases are countered by just around one hundred deubiquitinases many of which appear to show little regulation and specificity. Unraveling how deubiquitination is regulated is an exciting field with a high degree of medical relevance. The authors look into the regulation of UCHL3 by phosphorylation and utilize a wide range of techniques from biochemical activity assays to crystal structures. The paper does not provide conclusive results and ignores relevant papers on the subject. A small conformational change is observed and a qualitative activity assay is presented, which, due to inherent issues discussed below, is not likely to measure activity changes correctly. The authors show that UCHL3 cannot cleave di-ubiquitin, confirming a prior result that should be cited. The mechanism behind UCHL3’s ability to bind to di-ubiquitin without cleaving it, while cleaving mono-ubiquitin has been described in detail (PMID: 33238157). The authors show that UCHL3 also fails to cleave longer ubiquitin chains, except for chains bound to cyclin, which are cleaved. Regulation of UCHL3 by phosphorylation is an important topic and the author’s approach is valid. The issues with the experiments described in more detail below need to be addressed and the paper needs to be refocused on regulation by phosphorylation, removing speculative sections on questions that have been addressed in the literature already.

Major issues:

The authors compare the crystal structure of UCHL3-S75E with the wild type structure (PDB: 1UCH) and find a small difference in the orientation of the helix surrounding the mutation. They conclude that the structural change is caused by the mutation, however the proteins were subjected to different buffer conditions and crystallized in different space groups. Crystal contacts can lead to substantial conformation changes, far exceeding what the authors have observed. The authors should provide an analysis of the crystal packing in the supplement and discuss the limitation of their approach, especially in case the helix of interest engages in a crystal contact in one of the structures, but not the other.

The authors claim that the phosphorylation mimic mutation increases the deubiquitination activity of UCHL3 by a factor of 2.5. They provide the results of a fluorescent assay that uses ubiquitin-AMC, a single ubiquitin moiety bound to 7-amino-4-methylcoumarin. Cleavage of the ubiquitin C-terminus liberates the AMC unit and increases fluorescence. They present in Fig. 3 a plot of reaction time vs. fluorescence that shows a close to linear increase of fluorescence followed by an asymptotic region, suggesting that they have followed the reaction until the substrate was completely exhausted. Why is the final fluorescence for UCHL3S75E twice as high as the fluorescence measured with the wild type? Even if UCHL3S75E cleaves ubiquitin twice as efficiently, both reactions should reach the same final level of fluorescence if the same amount of substrate was there in the beginning. Trivial pipetting errors aside, this might point to general issues with the stability of the enzyme during the assay. It might be beneficial to perform a small buffer screen in order to find a better condition that allows correct measurements to be taken. I am a bit surprised by the absence of salt in the buffer. Providing 1mg/ml ovalbumin is good, I have had good results with BSA too. Maybe the most promising parameters to screen here would be detergents, Brij-35, Tween 20 or NP40 can help keeping enzymes at nanomolar concentration in solution. If the authors succeed in increasing the stability of their assay, they are encouraged to determine KM/Kcat in order to compare the enzymatic properties of their mutant in a quantitative manner.

A previous study (PMID: 25159004) that investigated cleavage of a rhodamine substrate (similar to the ubiquitin-AMC used in this study) and di-ubiquitin of various linkage found that “seven DUBs including all members of the UCH family as well as OTUD6A, JOSD1 and JOSD2 were active in the Rhodamine assay but not in the MALDI-TOF DUB assay.” This study should be cited by the authors. The authors came to the same conclusion based on their experiments where they mixed UCHL3 WT and mutant with di-ubiquitins of various linkage, incubated for up to 10 hours and subsequently analyzed the reaction by SDS-PAGE. The gel images in figures 4 (and 5) are very obviously spliced together from lanes taken from separate individual gels. While it is acceptable to assemble figures from gel lanes of different gels, there must be a blank border between them to ensure the reader understands that it is a composite figure. Figures 4 and 5 need to be changed accordingly. The bands need to be labeled. No positive control is shown, so whether the di-ubiquitin used in the assay can be cleaved at all is unclear. While this may be seen as less of a concern when well-characterized commercial substrates are used, the substrates in this study were obtained from an academic laboratory and their synthesis and the steps taken to ensure quality control were not described. It would be helpful to provide at least a reference to the synthesis of the substrates. Why was no reducing agent used during the reaction, especially when the reaction is catalyzed by a cysteine?

Whether tri-ubiquitin chains are indeed cleaved in Fig 5E is not entirely clear as the di-ubiquitin bands are also visible in the controls, albeit a bit weaker. Since control and experiment were analyzed on different gels (and sliced together in the figure), staining could be different. If the authors want to make this point, they need to repeat this experiment with control and reaction on the same gel and try to quantify the bands by densitometry. 

The authors show evidence for the cleavage of tetra-ubiquitin chains from cyclin. Unfortunately, the first sample was taken after 6 hours when the reaction had already run to completion. To compare the kinetics that the authors insist is altered by phosphorylation, they should analyze earlier steps. What might help here is the ‘blue silver’ staining method (PMID: 15174055), a Coomassie stain variant that is very sensitive and quantitative and may allow to take more samples during the reaction without increasing the use of valuable materials.

Author Response

(The authors gave the same response as above.)

Round 2

Reviewer 3 Report

I am grateful for the detailed rebuttal prepared by the authors. The authors have addressed some, but not all of the issues.

1.)   Interpretation of the crystal structure: I agree with the authors that the conformation change observed is likely a product of the mutation. The authors should state whether the loop in question forms a crystal contact in one or both structures and add a figure showing the crystal contacts in the wild type and mutant enzyme crystals.

2.)   Fluorophore assay: The authors need to address the enzyme stability issues that are very obvious from the plot they present. At present, the difference in activity could simply be a difference in enzyme stability, with the mutant losing activity faster in the assay buffer used. The authors need to find a better buffer system that preserves the activity of both enzymes. They should add a control measurement with the fluorophore by itself at substrate concentration so the degree to which the substrate has been cleaved can be determined. A comparison of Km/Kcat would be highly recommended. For this experiment, when using fluorescent assays, competition assays with mixtures of unlabeled and labeled substrate are a viable approach.

3.)   SDS-PAGE assays: The authors have addressed all issues with the gels and I would encourage them to provide the original gel images as supplementary figures.

Round 3

Reviewer 3 Report

I commend that the authors have taken the time to address all issues. The activity assay has greatly benefited from their screening efforts to identify suitable detergents and consequently the activity assay plots are now clearly interpretable and can be published in this form.